# Language Models as Critical Thinking Tools:
# A Case Study of Philosophers

**Andre Ye**[αγ]**, Jared Moore**[β]**, Rose Novick**[γ]**, Amy X. Zhang**[α]

andreye@uw.edu, jlcmoore@stanford.edu, amnovick@uw.edu, axz@cs.washington.edu
[α] Paul G. Allen School of Computer Science and Engineering, University of Washington
[β] Department of Computer Science, Stanford University
[γ] Department of Philosophy, University of Washington

## Abstract

Current work in language models (LMs) helps us speed up or even skip thinking by accelerating and automating cognitive work. But can LMs help us with *critical thinking* — thinking in deeper, more reflective ways which challenge assumptions, clarify ideas, and engineer new concepts? We treat philosophy as a case study in critical thinking, and interview 21 professional philosophers about how they engage in critical thinking and on their experiences with LMs. We find that philosophers do not find LMs to be useful because they lack a sense of selfhood (memory, beliefs, consistency) and initiative (curiosity, proactivity). We propose the *selfhood-initiative* model for critical thinking tools to characterize this gap. Using the model, we formulate three roles LMs could play as critical thinking tools: the Interlocutor, the Monitor, and the Respondent. We hope that our work inspires LM researchers to further develop LMs as critical thinking tools and philosophers, and other 'critical thinkers' to imagine intellectually substantive uses of LMs.

## 1 Introduction

> ***"But I like the inconveniences."*** — "We don't," responds the Controller. "We prefer to do things comfortably." — "But I don't want comfort," John gasps. "I want God, I want poetry, I want real danger, I want freedom, I want goodness. I want sin." — "In fact," says the Controller, "you're claiming the right to be unhappy ... the right to live in constant apprehension of what may happen tomorrow; ... the right to be tortured by unspeakable pains of every kind." There is a long silence. "I claim them all," says John at last. (*Minimally adapted from Huxley (1932).*)

Language Models (LMs) have recently alleviated a whole host of our intellectual inconveniences. They can help us do things we would have begrudgingly done by ourselves otherwise: write code (Chen et al., 2021; Rozière et al., 2023), generate emails (Goodman et al., 2022), and translate text (Costa-jussà et al., 2022). In sparking ideas by generating stories (Schwitzgebel et al., 2023) and concept designs (Cai et al., 2023), LMs offer shortcuts to gaining new thoughts. They also help us put our thinking into words by revising (Mysore et al., 2023) and giving feedback (Liang et al., 2024) on our writing. In all these cases, LMs help us speed up and circumvent the inconveniences of thinking ourselves.

In many contexts, however, the "inconvenience" of thinking is not a temporary problem to be alleviated, but a deep puzzle to be reflected upon. Many people are invested in specific areas of intellectual inquiry — e.g., historians, scientists, philosophers — and more generally, in reflection and engagement with the world — e.g., as informed political citizens, critical information consumers, and moral actors. They are interested in identifying and challenging assumptions, clarifying muddled ideas, and engineering new and useful ways to think. Core to this sort of inquiry is *critical thinking* — "the propensity and skill to engage in an activity with reflective skepticism" (McPeck, 2016). Can LMs serve as tools for *critical*

*thinking* — helping us think more deeply and in more complex ways, rather than faster or not at all? What if — like John — *we claim all the rights to think* (Buçinca et al., 2021)?

To investigate how LMs can serve as critical thinking tools, we use philosophers as a case study — philosophers being people who are in the business of thinking critically about a wide range of concepts and ideas. We interview **21 professional philosophers** to understand their thinking processes, collect their experiences with and views on current LMs, and brainstorm the roles LMs could play as critical thinking tools in philosophy (§3). We find that current philosophers *do not* think LMs are good critical thinking tools (§4) for two primary reasons: LMs are too neutral, detached, and nonjudgmental (§4.2); and LMs are too servile, passive, and incurious (§4.3). We propose the *selfhood-initiative* model for critical thinking tools, which explains why philosophers find conversations with other philosophers and reading philosophical texts to be more helpful for their work than current LMs (§5.1). Using the model, we describe **three roles** LMs could play as critical thinking tools: the Interlocutor, the Monitor, and the Respondent (§5.2). Finally, we outline how these LMs could inform metaphilosophical questions and shape the discipline of philosophy (§6.3), and discuss challenges in building LMs (§6.1) and interfaces (§6.2) for critical thinking.

## 2 Background and Related Work

### 2.1 Language Models as Thinking Tools

A large and growing literature investigates how LMs can serve as thinking tools for humans engaged in intellectual work. This research tends to concern how LMs can serve two intellectual functions: *idea stimulation* (roughly, "divergent thinking") and *idea refinement* (roughly, "convergent thinking") (Banathy, 1996; Design Council, 2019).

LMs can provide *stimulus for ideas* — information and (re)formulations which provoke and guide creative processes. Generally, LMs can expand idea sets (Fede et al., 2022), produce creative analogies (Bhavya et al., 2023) and metaphors (Chakrabarty et al., 2021); discover concepts (Lam et al., 2024), and facilitate group ideation (Rayan et al., 2024; Shaer et al., 2024). LMs to open-endedly propose plots, characters, and entire stories for creative writers (Calderwood et al., 2020; Schmitt & Buschek, 2021; Yuan et al., 2022; Mirowski et al., 2023; Chung et al., 2022; Chakrabarty et al., 2023); but also provide inspiration in more constrained creativity tasks, such as science writing (Gero et al., 2021; Kim et al., 2023). Although fraught with pitfalls (Messeri & Crockett, 2024), scientists can use LMs to find and synthesize literature (Van Dinter et al., 2021; Wagner et al., 2022; Fok et al., 2023; Khraisha et al., 2024) and iterate through research inquiry (Wang et al., 2023a; Morris, 2023; Liu et al., 2024). Designers can use LMs to generate and develop concept designs (Cai et al., 2023; Liu et al., 2023; Ma et al., 2023; Chong et al., 2024; Ma et al., 2024; Chen et al., 2024).

On the other hand, LMs can also aid the *refinement of ideas* – selecting from and improving upon an established existing pool of ideas. LMs can help writers by making suggested revisions (Du et al., 2022; Zhao, 2022; Mysore et al., 2023; Shu et al., 2023) and clarifying writing goals (Arnold et al., 2021; Kim et al., 2024). For scientists, LMs can facilitate revision of scientific writing (Liang et al., 2024; Radensky et al., 2024); for designers, LMs can provide feedback on (Duan et al., 2024) and annotate (Lu et al., 2024) UIs. In teaching settings, writing feedback given by LMs may be more motivating (Meyer et al., 2024) and engaging (Tanwar et al., 2024) than feedback given by other humans. Besides reviewing ideas, LMs can also curate them — for instance, by summarizing writing (Fabbri et al., 2021; Dang et al., 2022) and identifying important ideas (Lin et al., 2024).

### 2.2 Language Models as Critical Thinking Tools

However, one part of the thinking process is clearly missing. One does not simply go from the stimulus for ideas to figuring out how to refine them: one needs to do the actual *critical thinking*, involving reflection upon ideas, judgment, and conceptual engineering. LMs can help provide the seeds for our ideas when we don't have any (i.e., stimulus) and help us

work through them once we've got them (i.e., refinement), but how can they help us with questioning, reorienting, analyzing, and developing ideas (i.e., critical thinking)?

There are many different definitions of critical thinking: "the propensity and skill to engage in an activity with reflective skepticism" (McPeck, 2016), "reasonable, reflective thinking that is focused on deciding what to believe or do" (Ennis, 1993), and "the development and evaluation of arguments" (Facione, 1984), among many others. Critical thinking requires many dispositions, such as seeking clear statements of questions, looking for alternatives, and being open-minded (Ennis, 1987). Critical thinking is what makes many areas of intellectual inquiry — such as history, science, and philosophy — difficult. In these areas, people must produce and work with observations that are incomplete and open to a multiplicity of framings to pursue problems with often unclear definitions of progress — a landscape which demands critical thinking. For instance, on different accounts, history requires interpreting the past with alternative (nonlinear, long-range) temporalities (Braudel, 2023), taking into account the ways in which power structures shape historical record and memory (Foucault, 1969a; Trouillot, 1995), and identifying and manipulating narrative structures (White, 1973; Gaddis, 2004). Science requires advances not only in empirical work, but also reflection upon underlying paradigms of research (Kuhn & Hawkins, 1963), epistemology (Harding, 2013), and the social and material factors that constitute scientific knowledge (Latour, 1989).

Researchers across a variety of fields have developed a rich tapestry of approaches and tools to support critical thinking and related acts. Educators develop teaching strategies to promote critical thinking (McPeck, 1990; Pithers & Soden, 2000) such as teaching and interlinking a variety of perspectives on a subject in an integrative manner (Enciso et al., 2017) and encouraging students' intellectual independence in finding answers to their questions (Langer, 1997; Raths et al., 1966). Psychologists and cognitive scientists seek to understand how cognitive mechanisms and biases inform how humans (should) develop critical thinking (Carey, 1986; Reif, 2008), emphasizing the cultivation of basic metacognitive "building blocks" of critical thinking (Pasquinelli et al., 2021) and teaching for "practical theory" (Gelder, 2005). Meanwhile, human-computer interaction (HCI) researchers explore how interactions with computer applications can facilitate critical thinking: designers can provoke experiences of discomfort (Benford et al., 2012; Halbert & Nathan, 2015); emphase understanding over rote expression in social contexts (Kriplean et al., 2012; Sun et al., 2017); and build small "nudges" (Liao & Wang, 2022) into interfaces which "prime" (Yamamoto & Yamamoto, 2018) users towards reflective critical thinking (Bentvelzen et al., 2022); among many others. Many of these themes will be revisited in our discussion of design proposals for LMs as critical thinking tools (§5.2).

A growing body of work has explored how LMs might contribute towards critical thinking. LM-based news and media can positively affect users' willingness to think through opposing or novel viewpoints, which can be applied to combat polarization and extremism (Tanprasert et al., 2024; Zarouali et al., 2021; Shin, 2022; Wang & Tanes-Ehle, 2022; Blasiak et al., 2021). Cai et al. (2024) consider how currently "sycophantic", "servile", and "lobotimized" LMs can be used in more critical ways by challenging users' pre-existing ideas and constructively using antagonistic interactions to develop their thinking. Danry et al. (2023); Ma et al. (2023); Park & Kulkarni (2023) show how LMs can facilitate human self-reflection and improve human reasoning by asking questions instead of only answering them (as in the typical LM interaction paradigm). Xu et al. (2024) encourage critical thinking by building LM interactions using structured templates (over free-form chat). In more targeted contexts, LMs can be used to help scientific researchers critically think about their impact statements (Mukherjee et al., 2023), and to help political theorists to metacognitively reflect upon their own creative processes and judgments (Rodman, 2023).

## 2.3  Philosophy as Critical Thinking, Critical Thinking as Philosophy

In this paper, we focus on philosophy as a case study for critical thinking. Philosophy is concerned with critical, systematic, and reflective examination of the world. This includes understanding the basic structure of life and the world — what does it mean to exist (Aristotle, 350 BCE; Heidegger, 1927; Sartre, 1943), live (Aurelius, 180 AD), and die (Kierkegaard, 1983; Nietzsche, 1892)?; what does it mean to know something (Plato, 369BCE; Kant, 1781;

Husserl, 1931) and what are the limits of scientific knowledge (Popper, 2002; Chalmers, 2013)?; on what moral bases should we act (Aristotle, 350BCE; Spinoza, 1677), and is it even possible to determine 'objective' answers to moral questions (Hume, 1739; Harman & Thomson, 1996; Foucault, 1976)? Core to philosophy is "the endeavour to know how and to what extent it might be possible to think differently, instead of legitimating what is already known" (Foucault, 1976). Philosophy is for intellectual creation and engineering: Deleuze & Guattari (1991) wrote that "So long as there is a place for creating concepts, the operation that undertakes this will always be called philosophy." In thinking about how to think, philosophy is not only about *suspicion* toward the meanings and functions of phenomena, but also *recovery* of new significances and coherence (Ricoeur, 1981).

Contrary to the image that philosophy is "done in the armchair", isolated and impractical, philosophy has always been intertwined with other lines of inquiry. Plato engaged extensively with advanced mathematics; Aristotle contributed to early physics; Hume leaned on psychology. Philosophy has asked and continues to ask urgent, relevant questions: for instance, how are we to understand the strangeness of quantum mechanics in physics (Carnap, 1966); the relationship between consciousness (mind) and the brain (matter) (Chalmers, 2013); and "fairness" and "justice" in contexts like algorithmic discrimination (Hu, 2023), legal punishment (Alexander, 1922), and the distribution of resources (Rawls, 1971)? Indeed, researchers in every area of intellectual inquiry confront philosophical questions in their work: they might ask if a model or concept is "really real" (and how they know so), what the "nature" of their object of study is, aim to formulate normative desiderata for their theories, and so on. Therefore, we study philosophers' views and practices in this paper both because philosophers engage extensively in critical thinking *and* because many questions which require critical thinking asked by non-philosophers often have a philosophical flavor.

## 3   Methods

The first author conducted interviews with 21 professional philosophers at 14 philosophy departments at doctoral universities in the United States. We contacted and selected philosophers for high diversity across area of interest (e.g., ethics, political philosophy, philosophy of science). Interviews took place online and lasted between 30 to 60 minutes, depending on interviewee availability. Interviewees were asked how they philosophize (e.g., where ideas come from, how ideas are developed, what resources are needed) and their views on LMs (e.g., can LMs 'do' philosophy, how might they be useful for philosophizing). These questions followed a loose script (see §B), although we asked novel follow-up questions to pursue interesting lines of inquiry raised by the interviewees' responses. In cases where interviewees had very little or no prior exposure to LMs, they interacted live with the GPT-4 model on a philosophical topic of their choosing. We received IRB approval from our university to conduct the interviews; all interviewees confirmed their consent to participate in the study, and for their responses to inform the development of this paper. We qualitatively analyzed interview recordings and transcripts. Using an inductive approach (Thomas, 2006) and open coding (Charmaz, 2006), we identified common themes and positions (yielding §4 and §5). We refer to interviewees with a unique identifier, e.g., (P1, P2, P3) (see §A).

## 4   Language Models Are Not Good Critical Thinking Tools (So Far)

Many of the interviewed philosophers find LMs to be relevant and interesting, and some find them to have limited uses such as for undergraduate instruction (P1, P13, P20) or becoming acquainted with a topic (P5, P11, P12). However, none of the philosophers were convinced that current LMs can reliably and conveniently assist them in the intellectually substantive ways which require critical thinking. Philosophers described current LMs as "boring" (P2), "anodyne" (P4), "bland" (P9), and "cowardly" (P13). We discovered two broad reasons for this. First, current LMs tend to be highly neutral, detached, and non-judgmental, often commenting on ideas in abstract and decontextualized ways (§4.2). Second, current LMs tend to be servile, passive, and incurious, which is unhelpful when the user does not yet have a clear vision of what they want to accomplish, restricting the variety of intellectual interactions possible S4.3).

## 4.1 How do philosophers philosophize?

A close investigation of how philosophers think through difficult philosophical questions can give us insight into the types of tools and interactions which support difficult critical thinking, and provide contrast with current LMs, which fail to perform the same function.

*Where do philosophical ideas come from?* Philosophers report that their ideas usually come from observing puzzles and tensions in the world, in which some aspect feels bothersome (P5, P12, P20), incomplete (P10, P14), in need of clarity (P1, P13), or outright incorrect (P3). Philosophers encounter these puzzles and tensions most commonly in open conversation with others (P1, P2, P5, P9, P19) and while reading texts — books, papers, and monographs making explicitly philosophical arguments or touching upon philosophical themes (P4, P7, P10, P12, P13, P20). These puzzles may have an intellectual or logical character: terms might not be sufficiently disambiguated, inferences may not be valid, and propositions may entail absurd conclusions (P8, P11). However, for many, these tensions are identified and drawn out by ethical motivations (P1, P8, P16, P12). Tensions might arise not primarily because a proposition is incoherent, but rather because it appears ethically problematic. For instance, the trolley problem dilemma was used to probe the differences between doing and allowing harm, with applications to bioethics, particularly abortion (Foot, 1967). Several philosophers describe being inspired by texts communicating empirical work, seeking to provide explanations for empirical observations (P1, P2, P16, P18) as well as subjecting the practices and products of the empirical sciences to critical inquiry (P2, P7, P12, P13, P18).

*What do philosophers want out of their ideas?* Once philosophers identify puzzles from conversations and texts, they aim to develop ideas which make progress on these puzzles. Progress is conceived of in many ways: *"understand[ing] some part of the world better"* (P3), working through new ways to think about problems (P17), and better understanding the current ways we think — for instance, by making implicit assumptions explicit and recognizing the implications of propositions (P7). Some philosophers describe a developed philosophical idea as a "picture" (P9, P10) which organizes subideas in a systematic way, allowing one to clearly see the main point(s). This often requires "conceptual engineering" (P6): challenging, disassembling, and rebuilding the ways in which we think.

*The role of texts in philosophical development.* Texts continue to actively support the philosophical development past the inception of the idea. Revisiting texts with an idea in mind can unearth new aspects of the text which comment on that idea (P9), and repeatedly consulting written ideas can be helpful for putting words to newly developed ideas (P2, P20). Because texts are static and highly accessible by many people, texts can become a shared basis for and markers in conversation with others (P9, P19). Moreover, because published texts are usually produced by people who have given a problem substantial time and thought, philosophers might approach them with more trust and charity (P4).

*The role of conversation in philosophical development.* Conversations with fellow philosophers are central to evaluating the coherence of ideas (P21), raising connections to other ideas and problems (P5), and collecting feedback (P3, P10). Conversations may force philosophers to explain and justify ideas they may have taken for granted (P1). Conversation helps philosophers gain confidence that their ideas are good intellectual contributions (P2, P21). Philosophers even simulate conversations in their head, taking on various positions for and against their ideas (P1, P12). Good philosophical conversation requires several conditions. The interlocutor should be charitable — genuinely listening to and working through ideas (P1, P12), and trusting (P6, P14) — but also willing to boldly push ideas forward (P3) and take intellectual risks (P18). Conversations may not be directed towards any clear goal; interlocutors must be able to *"riff off each other"* (P8) and be willing to operate without a preset agenda (P3, P4). This requires interlocutors to be curious about addressing problems (P21); it should be a collaborative effort, rather than a combative debate (P3, P7).

## 4.2 Language Models are neutral, detached, and nonjudgmental

Philosophers find intellectual value when the conversations and texts they encounter provide substantive and well-defended perspectives, but find that LMs do not do the same.

**①** *LMs are abstract, imprecise, and 'skirt by' questions.* Because philosophy is interested in clearly stating and reflecting upon ideas, philosophers often place high value on precision in language. Changes to a formulation which seem trivial to a layperson may introduce important shifts in meaning for a philosopher. Meanwhile, LMs seem as if they 'tell the user what they want to hear', resulting in risk-averse and hand-waving behavior which produced abstract, imprecise, and ultimately intellectually uninteresting statements (P5, P7, P15). Interviewees noted that when they brought up problems with LMs' responses, LMs skirted around the issue, producing superficially convincing corrections without really addressing the provided issue (P1, P20). LMs are highly factually knowledgeable (P1) but fail to precisely express philosophical ideas; thus, LMs end up reinforcing the status quo rather than proposing substantive and interesting challenges (P9).

**②** *LM responses change too easily and don't have 'weight'.* Several philosophers describe how easy it is for them to talk LMs into contradictions and incoherent outputs in the same session (P4, P9). LMs make "kneejerk reactions" to user concerns and are excellent at effusively apologizing, but don't "*fully appreciate*" their mistakes and the user's comments (P14). Moreover, LM responses seem highly sensitive to trivial changes in the prompt, making some philosophers wary of using them at all (P21). The ease with which one can manipulate an LM's output seems to reduce their trustworthiness and value as tools (P15).

**③** *LM outputs don't provide judgments.* LMs often refrain from formulating serious judgments; they try to remain neutral and 'see all sides', but end up presenting all sides in placid and uninteresting ways (P12, P17). They tend to refrain from discussing controversial issues (P4), which is unfortunate given that philosophy prides itself on clearly thinking about otherwise-taboo topics of controversy. As such, LMs are perceived as "*cowardly*", refusing to take solid positions and, in some sense, echoing the user (P13). *"It [conversations with LMs] ends up being unproductive and unsatisfying... they don't feel like persons because their language is often so bland and impersonal, non-Socratic, generic... they're boring"* (P9).

**④** *LMs don't have memory and context.* Shared context from previous interactions with other humans serve to provide context for and situate ideas in conversation, allowing for efficiency of exploration (as already-exhausted ideas are not brought up again) (P1, P14). Because current popular LM interfaces 'lose their memory' of previous interactions in different sessions, LMs often produce general and decontextualized responses to user prompts (P15).

### 4.3 Language Models are servile, passive, and incurious

Philosophers find intellectual value when fellow philosophers develop their own lines of inquiry in conversation and texts, but find that LMs do not do the same.

**①** *LMs fail to be useful in open, undetermined contexts.* LMs enthusiastically make "*my problem its problem*" (P11), but often philosophers do not have their 'problem' entirely clearly thought or formulated (P5). For certain basic tasks, "*'you have certain success metrics in mind, so you go to [an LM]; but what about truly open-ended conversations where you don't have success conditions already laid out?"* (P7) LM answers often feel like they've been 'packaged' or return a 'processed end result', whereas *"in the doing of philosophy, we want to be open, in service of a larger dialogue — philosophy as a process rather than as an end product"* (P5). LMs don't seem to have a drive to know the truth or care about convincing people (P2, P21) — features which interviewees note energize interactions even when there is no clearly desired product.

**②** *LMs restrict the variety of intellectual interaction.* The "incuriosity" of LMs severely limits possible intellectual interactions philosophers can have with it (P7). *"It's a question-answer platform. It won't follow up with a "what do you think?" "I'm a little puzzled, how it could be?" "Oh gosh, how does it work?" You can't have a conversation with [an LM] except one which is like an interview."* Several philosophers imagine alternative useful LM interactions in which LMs take on more intellectual risks and independent behaviors: instead of only answering questions, LMs could also ask them (P12, P17), or LMs might behave with hostility and antagonism towards users' ideas (P6, P8, P11).

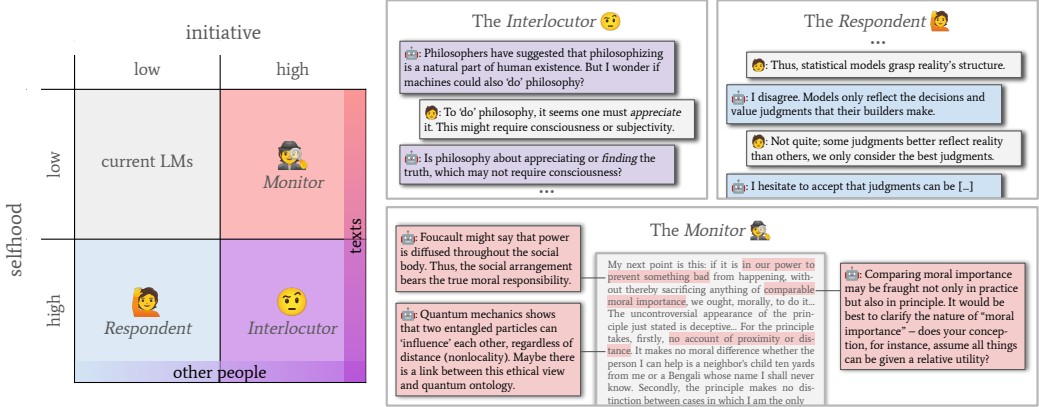

Figure 1: *Left* – The selfhood-initiative model for critical thinking tools. *Right* – Illustrative sample interactions between humans and LMs playing different roles. Other alternatives are possible. The excerpt from "The Monitor" is taken from Singer (1972).

# 5   Designing Language Models for Critical Thinking

Thus far, we've introduced the problem of critical thinking and described how current LMs fail to be good critical thinking tools for philosophers. Here, we set out a formal model to characterize and compare critical thinking tools (§5.1). This allows us to imagine new roles for LMs, inspired by what makes people and texts useful as critical thinking tools (§5.2).

## 5.1   The Selfhood-Initiative Model

We use the two broad reasons why LMs fail to be good critical thinking tools in §4 as the basis for the model's two axes: current LMs have low *selfhood*, as they are neutral, detached, and nonjudgmental; they have low *initiative*, as they are servile, passive, and incurious. In particular, *selfhood* is a resource's ability to have certain locally persistent internal states (such as perspectives, beliefs, opinions, memory) and to consistently use them as the basis for judgments. The resource's internal states may change over time due to new knowledge and experiences, but in an intentional and logical (rather than an arbitrary and capricious) manner. Current LMs exhibit *low selfhood* (§4.2). *Initiative* is a resource's ability to set its own intentions and goals, possibly different from its user's, and to execute actions oriented towards those intentions. High-initiative resources are not strictly or existentially bound to their user's directives, and may deviate from them. Current LMs exhibit *low initiative* (§4.3). These two axes form the *selfhood-initiative* model for critical thinking tools. Our model is distinct from previous models proposed for the study of critical thinking in that (a) we model *types of critical thinking tools* rather than the *(human) process of critical thinking* (Schön, 1987; Shneiderman, 2000, *inter alia*), and (b) we explore the *interaction* between selfhood and initiative, which have each independently been explored in some capacity by others (Cai et al., 2024; Guo et al., 2024; Hilliard et al., 2024, *inter alia*). Our model explains why philosophers find texts and other people (but not LMs) to be useful tools, and further provides a design space for LMs as critical thinking tools (§5.2).

*Why do philosophers find other people and texts to be useful critical thinking tools?* In the selfhood-initiative model, other people are *high-selfhood, variable-initiative tools*. People have specific backgrounds and experiences which inform their views, perspectives, and beliefs; these influence how they understand and respond. Philosophers find value in talking to other people often *because* of their selfhood; they expect that they will receive interesting judgements and comments, rather than placid neutrality. However, these people may have variable initiative, depending on the situation. In free-flowing conversation, each interlocutor may carry the conversation in some direction, whereas in a more focused conversation aimed at collecting feedback, an interlocutor may be expected to directly respond to one's ideas and requests without their own intellectual initiative. The high selfhood of other people is

helpful because it provides particular perspectives and ways of looking into the problem space. Meanwhile, in the selfhood-initiative model, texts are *high-initiative, variable-selfhood tools*. Texts are not themselves responsive to a user's intentions (Plato, 370BCE); they express the author's attempt to fulfill their intentions, and one encounters the product of this attempt after the fact of its production. The text's exteriority from the user allows the user to reflect upon similarities and differences between their own thinking and the tool's outputs. On the other hand, the way in which texts are written can vary in the degree of selfhood they express. Informative, survey-based, and clarificatory papers tend to de-emphasize an author's perspectives and opinions, whereas more explicitly argumentative papers may center them; both can be useful to philosophers in different ways.

*Why don't philosophers find current LMs to be useful critical thinking tools?* In the selfhood-initiative model, current LMs are *low-selfhood, low-initiative tools*. They do not provide philosophers with particular concrete perspectives into the problem space, nor do they provide ideas sufficiently exterior to a philosopher's own thinking to allow for meaningful reflection and connections. These properties make LMs particularly useful for alternative modes of thought, such as carrying out rote and well-defined tasks and helping rewrite sentences, but not for stimulating critical thinking.

## 5.2 Three Roles for Language Models as Critical Thinking Tools

According to the selfhood-initiative model, good critical thinking tools should have high selfhood, high initiative, or both. From our model, we set out *three roles* of LMs for philosophy — the Interlocutor, the Monitor, and the Respondent — corresponding to the *three viable cells* in the selfhood-initiative model (high-selfhood, high-initiative; low-selfhood, high-initiative; high-selfhood, low-initiative). Implementations for these roles vary — some might be achievable with only moderate prompt engineering, whereas others might necessitate radically different user interfaces or model training methods.

*The Interlocutor ∘ high-selfhood, high-initiative*. Philosophers mention that they often get their ideas in free-flowing conversation with fellow philosophers or from reading literature that makes arguments which seem tenuous, incorrect, or incomplete (§4.1). In the terms of the selfhood-initiative model, these are *high-selfhood, high-initiative* tools. As a role for LMs, the Interlocutor would invert many of the human-AI relationships taken for granted in current LMs. Rather than attempting to remain neutral, the Interlocutor makes judgments and takes positions based on its perspectives. Rather than accommodating and affirming users' every response, the Interlocutor thinks through and challenges or disagrees with what its users say; it responds or modifies its own beliefs if users make reasonable points. Rather than remaining passive and answering user questions, the Interlocutor asks its own questions in pursuit of its 'own' interests, and refuses or redirects certain lines of inquiry in favor of others. Rather than being amnesic and detached, the Interlocutor draws upon its persistent memories and beliefs across sessions to produce ideas. The Interlocutor does not need to be strictly *antagonistic*, as explored in Cai et al. (2024); indeed, it may be charitable and polite, much like colleagues, while at the same time resisting the 'servility' and 'sycophancy' disrupted by the antagonistic paradigm.

*The Monitor ∘ low-selfhood, high-initiative*. While developing ideas, philosophers consciously or unconsciously encounter various "*decision junctures*" at which they use certain approaches or pursue certain ideas over others (P6). Many philosophers suggest that it may be important to reduce, or at least become more aware of, the choices at 'unconscious decision junctures' (P6, P2, P7). Without such awareness, philosophers may expose their ideas to imprecision ('which path did you exactly take?') and objections ('why this path and not others?'); moreover, these choices may reproduce personal and disciplinary biases, reifying metaphilosophical problems (§6.3). As a role for LMs, the Monitor acts as a 'checks and balances' on philosophizing; it is not interested in retaining self-consistency or in expressing particular points of view (low selfhood), but has high initiative to provide a variety of ideas and resources to the user. The Monitor functions similarly to survey texts which provide a 'lay of the land', illustrating different approaches and ideas to help philosophers situate their ideas, able to take all sorts of changing sides with the initiative to challenge and confront. The Monitor's suggestions may or may not be directly relevant to the philosopher's work,

but act as reference guides — to which the philosopher might think, "that's a related idea, maybe there's a connection here" or "that doesn't seem directly related, but it's good to have in mind". Moreover, the Monitor may ask a variety of uncomfortable and unexpected methodological questions aimed at clarifying philosophers' decisions.

*The Respondent ∘ high-selfhood, low-initiative.* As philosophers develop their ideas, they want to understand how others might react — better understanding possible misinterpretations, objections, and clarification questions which may arise (P6, P10, P12). These reactions should have high selfhood to be substantive and particular, and low initiative to remain directly focused on the user's ideas. As a role for LMs, the Respondent adopts a specific set of beliefs and perspectives and reacts directly to the user's ideas; it does not merely role-play or superficially caricature different positions, but should have consistent memories and beliefs which are reasonably open to change (P4) rather than dogmatically fixed. Interactions with the Respondent may inform how the philosopher formulates and presents their ideas; they may anticipate certain objections and strengthen its appeal and utility. The Respondent can also be *counterfactually* helpful: if an agent representing an unsavory position resonates with a philosopher's argument, then that philosopher might reconsider how their argument is expressed, not only defending but also *delimiting* the scope of their argument (P6).

## 6 Discussion

### 6.1 Challenges for Language Modeling

Critical thinking can serve as another of many "north stars" in LM research, guiding what we want from LMs. Corresponding to the limitations of language models discussed in §4.2 and §4.3 are several concrete areas for further LM research. LMs will need to become more convincing agents (Andreas, 2022) which can represent specific positions and belief systems (Scherrer et al., 2023; Jin et al., 2024) ③; stay consistent with them (Chen et al., 2021; Zhao et al., 2024) ②; and commit towards and draw from long-term memory (Wang et al., 2023b) ④. In particular, LLMs will need to concretely reason about "uncommon sense" ① ②, seriously considering positions which deviate from intuitively true or correct ways of thinking about the world (Bisk et al., 2020; Ziems et al., 2023; Hendrycks et al., 2021; Pock et al., 2023). This may require rethinking how we align LMs (Ouyang et al., 2022; Sorensen et al., 2024), given that humans tend to be drawn towards confident common-sense responses (P5). LMs will need to improve their long-range planning (Hao et al., 2023) and act autonomously (Händler, 2023) ①, operating in cases where there is no clear algorithm for solving a problem (P4, P3, P8); LMs will need to take effective conceptual risks without clear immediate payoffs (P18) and reason about unsettled and open ideas (P8). To support more diverse forms of interaction beyond question answering or task execution ②, LMs will need to significantly improve in theory of mind (Jamali et al., 2023; Strachan et al., 2024). LMs need to "*understand what's happening [in the conversation] without it being explicitly said, because.. you haven't fully expressed it to yourself*" (P8), which will allow them to focus on the significant rather than irrelevant or obvious paths of inquiry in conversation (P6, P8).

### 6.2 Challenges for Human-AI Interaction

In addition to *modeling challenges*, there are several *interaction design challenges* when developing LMs for critical thinking. First, philosophers tend to highly value *thinking through things themselves*; many emphasize that the intellectually substantive parts of philosophy cannot be naively 'accelerated' (P1, P7, P14, P17). Philosophers find the process of thinking to be intrinsically valuable, even when it does not produce obvious payoffs (P3, P6, P8) — a feature common to other areas of critical thinking. Additionally, philosophers may feel that authorship of ideas requires that the ideas be '*mine*', and that '*I*' should be responsible for making the important intellectual judgments (P4, P10, P18). Secondly, *it can be difficult and even disruptive to put ideas into words*. Although professional philosophy is mainly formally done in language, the process of thinking through ideas can involve many other dimensions of representation and thinking (P2, P3, P4, P5). Among other challenges, philosophers cite the apparent incongruence between ideas and language as a source of significant burden

in learning how to effectively use LMs (P8, P21). This may be true for many other areas of critical thinking. Thirdly, philosophers find that *human connection is enjoyable and important*. Besides giving rise to unexpected philosophical connections and ideas (P6), conversation with another human is deeply enjoyable and fulfilling, on its own merits (P8, P21). Moreover, some philosophers feel that serious philosophical inquiry requires some kind of subjectivity or lived experience (P6, P8, P16). Therefore, LMs will need to coexist with and enrich, rather than seek to replace, the ecosystem of human and textual resources already available to philosophers and other professional critical thinkers.

### 6.3 LMs Help Think About and Address Metaphilosophical Problems

Throughout our interviews, we found that thinking through how LMs can serve as critical thinking tools raises many interesting metaphilosophical questions. What does it mean to 'do' philosophy, and who or what can 'do' it? How mechanical is philosophy? What is 'thinking'? Our findings in §4.1 provide some empirical illumination for these questions. Philosophers found concretely reflecting on these questions — provoked by thinking about LMs' role in doing philosophy — to be interesting and helpful (P1, P7, P15, P20).

However, LMs may also play an active precursory role in *addressing* metaphilosophical problems. Philosophers have articulated a host of concerns about the philosophical method and discipline: for instance, philosophers' standards for argumentation may exclude more diverse forms of philosophical inquiry Diamond (1982); Dotson (2012), and their methods for categorizing 'schools of thought' (such as the analytic-continental distinction) may be counterproductive (Dolcini, 2007), reconcilable (Levy, 2003; Bell et al., 2016), and not really substantive (Mizrahi & Dickinson, 2021; Thomson, 2019). Certainly, these concerns point towards deeply entrenched sociological features of the discipline. This entrenchment is a dialectic between disciplinary structure and individual philosophers, wherein the former (materially) constrains the latter and the latter works within the lines of (and reproduces) the former. LMs might contribute towards disrupting this second direction: drawing philosophers' attention outside the canon and across schools of thought as Interlocutors and Monitors, and representing these positions and methodologies as Respondents – possibly more approachably and accessibly than humans could. Consider Heidegger (1927)'s metaphorical carpenter: busy at work, the hammer is "ready-at-hand", unnoticed. It is when it breaks that it becomes "present-at-hand", noticed — an object of conscious reflection. Arguably, the philosopher must engage with ideas and methods present-, rather than ready-, at-hand (Plato, 380 BC), but the ability to engage in this way is a function of the tools and circumstances around us, and therefore often legitimately difficult (Ahmed, 2006). LMs can help, so to speak, 'make the present-at-hand, ready-at-hand' in a way that philosophical humans and texts cannot. Respectfully building LMs with selfhood and/or initiative into the philosopher's material workspace – the text editor, the article viewer, and so on – can prompt 'present-at-hand' reflection in quiet moments and directions which a philosopher may have neglected as ready-to-hand. These small interactions, at scale, might introduce cracks into metaphilosophical edifices that philosophers would like less entrenched.

## 7 Conclusion: Towards Living Script

In his masterwork *Jerusalem*, Moses Mendelssohn writes that philosophy has too long prioritized a dead form of interaction, one which stifles human interaction and innovation: "*We teach and instruct one another only through writings; we learn to know nature and human only from writings. We work and relax, edify and amuse ourselves through scribbling...*" (Mendelssohn, 1783, 41). In response, Mendelssohn calls for a turn towards a *living script*, "*arousing the mind and heart, full of meaning, continuously inspiring thought*". The living script is a way of engaging with tools that inspire and support our critical thinking; it is an ideal both for LM researchers, philosophers, and all of us — as thinkers and humans — to aspire towards. As potential technologies for reading and writing our living script, LMs can offer critical thinkers a more wide and accessible set of ways to support the development of ideas and to shape disciplinary practices and cultures. In the face of intellectual automation, it begins by saying, with John, for the rights and responsibilities to critically think: *"We claim them all."*

**Ethics Statement**

Although exploring 'uncommon sense' is important for critical thinking, we acknowledge that it can also be a deeply uncomfortable and unsettling experience. Disagreement can feel awkward in many contexts in daily life, even though it may not in designated spaces: *"one of the best gifts a philosopher can give another is a good counterexample... in philosophy, we like a challenge, a pushback, for people to think that we're wrong. That's where philosophers thrive"* (P5). Moreover, common sense encodes certain ethical or moral norms, such as "pain is bad" and "racism is unjust"; critical thinking tools may facilitate the revisiting and challenging of these norms in apparently inappropriate ways. To be sure, there is great value in this practice. We may not only want to believe in true things but also know the right or best reasons for *why* we should believe in them (in what sense of 'bad' is pain *bad*? *why* is racism unjust?), since having poor reasons for a belief may undermine the belief without our knowledge. Moreover, supposedly obvious moral principles and norms can be utilized to support positions we might think to be unsavory or misguided (e.g., racism is unjust, so we should only pursue a strictly 'colorblind' public policy); it is difficult to identify this if one does not adopt a critical view towards the entire system. Nevertheless, LMs can serve many purposes, and being critical thinking tools is just one of them. Low-selfhood and low-initiative tools are needed to accomplish many other important tasks. Users should consent to critical interactions with LMs.

Some interviewees expressed that LMs raised difficult questions about academic integrity and authorship of ideas. It should be noted that because critical thinking tools are intended to *support* the process of thinking rather than replacing it, there is little risk of outright *plagiarism*, provided the tools are designed properly and used as intended. Nevertheless, there are interesting ethical questions about ownership of ideas with respect to involvement in their development. If a colleague's offhand comment sparks an idea, leading to a publication, (how) should the colleague be credited? What if instead they intentionally discuss and develop an idea with you? What is an author (Foucault, 1969b)? The question of *how LMs as critical thinking tools should be credited* joins the broader existing rich discourse of how generative AI in general should be credited in intellectual production (Hullman et al., 2023; Jenkins & Lin, 2023; Simon et al., 2024; Springer, 2024, *inter alia*).

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

## A Interviewee Information Sheet

Table 1 provides high-level information about each interviewee which may be relevant to interpreting and contextualizing their views. The *General Interest(s)* feature describes the broad fields that the interviewees work in. The *Notable Specific Interest(s)* feature describes any specific topics in the field(s) mentioned in the *General Interest(s)* feature that the interviewees focus their work on. This feature is not exclusive, meaning that interviewees may also work on other topics outside of the specific interests. If the value for this feature is blank, then the interviewee's work is sufficiently characterized by the value in the *General Interest(s)* feature. The *Experience with LMs* feature describes three levels of experience with using LMs: little to none, limited, and extensive. If interviewees have either limited or extensive experience with using LMs, the *Uses of LMs* feature describes their primary use: for teaching (e.g., using LMs to teach material, trying to understand features of LM-generated student submissions), for personal use (e.g., to improve productivity, for entertainment), for exploration (i.e., playing around with the LM out of curiosity to understand the technology better), and for research (i.e., their research is on LMs). Note that the following interviewees have published at least one article on some aspect of AI: (P5, P6, P13, P14).

| ID | Title | General Interest(s) | Notable Specific Interest(s) | Experience with LMs | Use of LMs |
|---|---|---|---|---|---|
| P1 | Associate Professor | Ethics, Political Philosophy | Bioethics, Feminist Ethics | Limited | For teaching |
| P2 | Associate Professor | Philosophy of Science | Philosophy of Biology | Limited | For exploration |
| P3 | Professor | Ethics, Aesthetics | Meta-ethics | Limited | For teaching |
| P4 | Professor | Ethics, Political Philosophy | | Limited | For personal use |
| P5 | Assistant Professor | Ethics | Virtue ethics | Limited | For teaching |
| P6 | Assistant Professor | Ethics, Political Philosophy | Philosophy of Technology, AI | Extensive | For research |
| P7 | Assistant Professor | Philosophy of Science | Philosophy of Physics | Extensive | For personal use |
| P8 | Associate Professor | History of Philosophy | German philosophy | Limited | For personal use |
| P9 | Professor | Philosophy of Science | | Extensive | For exploration |
| P10 | Associate Professor | Ethics, History of Philosophy | Philosophy of Technology | Little to None | |
| P11 | Professor | Philosophy of Science | Philosophy of Statistics | Little to None | |
| P12 | Professor | Philosophy of Science | Psychology | Limited | For exploration |
| P13 | Associate Professor | Philosophy of Science | Philosophy of Biology | Limited | For class |
| P14 | Professor | Logic, Philosophy of Mind | Semantics, Linguistics | Extensive | For exploration |
| P15 | Assistant Professor | Aesthetics | Value theory, Literature | Limited | For exploration |
| P16 | Professor | Ethics, Political Philosophy | Public and Global Policy | Limited | For exploration |
| P17 | Teaching Professor | Pedagogy, Epistemology | | Extensive | For exploration |
| P18 | Associate Professor | Philosophy of Science | Philosophy of Physics | Limited | For exploration |
| P19 | Assistant Professor | Ethics | Moral psychology | Limited | For class |
| P20 | Professor | History of Philosophy | | Little to None | For personal use |
| P21 | Associate Professor | Ethics | Bioethics | Little to None | |

Table 1: Interviewee information.

## B   Interview Questions and Guidelines

1. Meta-philosophy
   (a) What is philosophy? Why do you go about doing philosophy? What aims do you have?
   (b) What drives the 'doing' of philosophy? What is the role of personal motivations, subjective experience, and aesthetic judgements?
   (c) Who or what can 'do' philosophy? For instance, can LLMs 'do' philosophy?
   (d) What makes doing philosophy 'difficult' / nontrivial?
   (e) How does philosophy distinguish its products from those of other disciplines?

2. The philosophical process
   (a) How do you go from no idea to a spark of an idea / an unrefined idea?
   (b) How do you develop and refine philosophical ideas? What moves have to happen?
   (c) How mechanical / creative is the process of doing philosophy?
   (d) What is the relationship between texts / textual methods and philosophy? Does philosophizing, to some extent, operate 'above' language in ideas / thoughts?
   (e) What is the role of conversation in the doing of philosophy? What are some of its challenges?
   (f) What makes for a good interlocutor, and what makes for a good conversation?

3. Language Models for philosophy
   (a) What roles can language models play in the development of philosophy?
   (b) What do language models need to be better in the development of philosophy?
   (c) What are some of the opportunities and strengths for language models in philosophy?
   (d) What are some of the risks and weaknesses for language models in philosophy?
   (e) Would you use language models in intellectually substantive ways currently? What about in the future, with plausible improvements?

