# OpenReview forum: "Language Models as Critical Thinking Tools: A Case Study of Philosophers"
_colmweb.org/COLM/2024/Conference — COLM_

### Official Review · Reviewer_veW6 · 2024-04-29

**Rating:** 5
**Confidence:** 3
**Ethics Flag:** 1

**Summary:**

While often LLMs are employed to quickly retrieve standard knowledge and arguments, this paper discusses the question to what extent LLMs can serve as tools for critical thinking in the sense of questioning, reorienting, analysing, and developing ideas. As a case study, the paper reports on interviews with 21 professional philosophers about how they engage in critical thinking and on their experiences with LMs. In view of the findings that LLMs are considered to fail to be useful in open, undetermined contexts, and also do not allow intellectual interaction driven by curiosity, the authors model conversations with LLMs along two dimensions – selfhood and initiative. Based on the resulting matrix of high and low selfhood and initiative, they propose three roles of LLMs for philosophy: the Interlocutor (high-selfhood, high-initiative), the Monitor (low-selfhood, high-initiative), and the Respondent (high-selfhood, low-initiative). In conclusion, they take up some suggestions of where LLMs might be improved (e.g. reasoning about “uncommon” sense, or improving their long-range planning), and advocate the use of LLMs in a playful way despite their insufficiencies.

**Questions To Authors:**

(1) By which criteria have the interviews philosophers been selected? Has it been ensured that the interviewed have experience with LLMs?
(2) Which LMs and Chatbots have been used?
(3) Have you been considering possible modifications of the experimental set-up?

**Reasons To Accept:**

The paper presents an interesting and novel classification of interactions with LLM based chatbots. While the findings of the interviews are not surprising, this classification helps to better understand the reasons behind the interviewed philosophers’ assessments.

**Reasons To Reject:**

The paper leaves out important details:
(1) By which criteria have the interviews philosophers been selected? Has it been ensured that the interviewed have experience with LLMs?
(2) Which LMs and Chatbots have been used?
(3) The claims are very general and do not take into account possible modifications of the experimental set-up. In particular, the noted insufficiencies may be a result of just the way how standard LLMs have been set up. An alternative might be databases of pro- and con-arguments that could be added by fine tuning or retrieval augmented generation (RAG) on such databases.
(4) From the philosopher’s side, the conclusion also appears too general, missing out differences in the intended gain of knowledge in the philosophical traditions and methods.

---

> ### Author Rebuttal · Authors · 2024-05-28
>
> We appreciate the reviewer’s feedback! To address the reasons to reject (RR) and questions:
>
> [Q1, RR1]: Philosophers were recruited from doctoral universities across the US, selected for variation across area of specialization (e.g., philosophy of science, political philosophy, ethics) and across experience with LMs. Some of our interviewees had published works on philosophical issues with LMs; others had only heard about LMs but not interacted with them. By interviewing a diverse set of philosophers, our findings more strongly reflect concerns shared across philosophers.
>
> [Q2, RR2]: In cases where philosophers did not have familiarity with LMs, we used a protocol in our interviews to let them interact with the GPT-4 model. However, most philosophers report having interacted with GPT 3.5/4, Claude, Bard, etc. Therefore, our findings are not restricted to one specific model.
>
> [Q3, RR3]: We agree with the reviewer’s suggestion that the noted deficits “may be a result of just the way how standard LMs have been set up”! Our paper does not aim to identify fundamental flaws with LM training, architectures, etc. Rather, we aim to document the experiences of philosophers with current mainstream LMs. Addressing some of these deficits may or may not be ‘simple fixes’, but there are good reasons to believe many are ‘deep-rooted’. Firstly, philosophers make clear that their concerns with current LMs are not only due to poor factual knowledge; the way in which LMs approach philosophers (without selfhood or initiative) makes LMs poor critical thinking tools – not entirely addressable with RAG. Moreover, many of the concerns are directly related to open and difficult research problems, such as building stateful & consistent LMs and skills in theory of mind (S6.2). Improving on these problems will help not just philosophers but critical thinkers in general.
>
> [RR4]: We agree that our discussion of philosophy is quite general. Philosophy is a very rich discipline with a wide diversity of methodologies, interests, and problems. Each subarea would use LMs in a very different way, and we agree that important future work would take a more narrow and concrete look at these uses. However, as we believe this paper is one of the first to introduce philosophers’ experiences with LMs into the LM literature, we want to present the general moves that philosophers make (described in S4.1) and experiences they have with LMs (S4.2-3) across these subareas.
>
> Thank you!

---

> > ### Comment · Reviewer_veW6 · 2024-06-04
> >
> > Dear authors,
> > thank you for your replies! I still think that your paper addresses two topics that are not so closely related as you presuppose. While I do see the merit of your contribution in the introduction of a novel classification of interactions with LLM based chatbots, I still have reservations on the methodology and experimental set-up of your survey of how philosophers interact with LLMs. Therefore, I'll keep my score.

---

### Official Review · Reviewer_NJTN · 2024-05-04

**Rating:** 10
**Confidence:** 5
**Ethics Flag:** 1

**Summary:**

This paper explores the potential for language models to serve as tools for critical thinking, using philosophy as a case study. The authors engage with prior work and conduct a sound qualitative study by interviewing 21 professional philosophers, providing valuable empirical insights into how philosophers think and their attitudes toward language models. The work demonstrates originality by examining language models for critical thinking rather than just text generation/editing assistance. The proposed "selfhood-initiative" model offers a practical conceptual framework, and the envisioned roles for language models like the Interlocutor, Monitor, and Respondent represent creative new possibilities. Significantly, the paper highlights significant gaps in current language models for supporting deeper reasoning required for critical thinking and articulates an agenda for developing better-suited models, which could have wide-ranging impacts. The discussion around language models shaping metaphilosophical questions and the discipline of philosophy itself is highly relevant. Overall, this is a remarkable work that makes a valuable contribution by studying an underexplored area at the intersection of AI and philosophy through a combination of good design, conceptual modeling, and creative proposals for future development.

**Questions To Authors:**

Did you try Claude? I have heard it's better than GPT4.

**Reasons To Accept:**

This is a high-quality paper. The authors know their topic well. The writing and organization are very impressive. The key questions and findings are helpful to the community, especially when there is a huge amount of anthropomorphizing of LLMs. LLMs are good at benchmarks, but that sort of reasoning often doesn't imply critical reflective thinking, and then there is the classic case of LLMs being servile, passive, and incurious (thanks to RLHF). This comes time and again in creative writing where writers feel that RLHF limits LLMs in exploring darker difficult topics. I appreciated the design and recommendations for what makes a better critical thinking tool. The SelfHood Initiative model is pretty exciting and will likely have a good impact on interdisciplinary researchers who work with LLMs

I would fight for this paper's acceptance.

**Reasons To Reject:**

No reason to reject

---

> ### Author Rebuttal · Authors · 2024-05-28
>
> We appreciate the reviewer’s feedback! In particular, we would like to concur with the reviewer’s argument that “LLMs are good at benchmarks, but that sort of reasoning often doesn't imply critical reflective thinking”. Increasingly, LM scientists are finding that the most useful metrics come not from static benchmarks but from real users, especially experts. We hope our paper, in addition to providing a detailed qualitative evaluation of a specific user population (philosophers), paves the way for future work on LM tools for philosophical and other critical thinking problems.
>
> Although in our interviews, we had participants interact with GPT-4 if they did not have familiarity with LMs, many of our participants had already interacted with a variety of LMs in the past, including GPT 3.5/4, Claude, Bard, etc. Therefore, our interviews document interactions with a variety of models. We would also note that Claude Opus had not yet come out during the time of the interviews.
>
> Thank you!

---

> > ### Comment · Reviewer_NJTN · 2024-06-05
> > **Thank you**
> >
> > Thanks for clarification. I will keep my score

---

### Official Review · Reviewer_Ynvx · 2024-05-11

**Rating:** 3
**Confidence:** 4
**Ethics Flag:** 1

**Summary:**

This paper is interested in the use of LLMs to assist humans in critical thinking. Academic philosophers, in interviews which are qualitatively coded, are asked to reflect on the use of LLMs to assist them in their work. Overall, the interviewed philosophers find that LLMs are not useful. The paper diagnoses this uselessness in two missing properties in LLMs: they lack a sense of selfhood and initiative. Possible roles for LLMs that vary these features (i.e., having selfhood and lacking initiative, lacking selfhood and having initiative, and having both) are discussed as a way of trying to guide the development of LLMs.

**Questions To Authors:**

Who is the “we” in “we claim all the rights to think?”. I read this as saying we humans claim all the rights to think. What relationship does that have to LLMs? It feels to me, in fact, like that statement is in tension with the aims of the paper (the description of a tool to automate some types of thinking). Perhaps, we will use LLMs to think more critically? But the other uses cases (e.g., coding, generating emails) mentioned in the beginning are taken as useful because they alleviate the need for that type of thinking.

Two issues with LLMs, that they are “highly neutral, detached, and non-judgmental” and “servile, passive, and incurious” are mentioned a few times in the paper, however, no empirical results are given to support this. If this is the perception of the interviewees that is fine, but it should be made clear in the paper.

**Reasons To Accept:**

I found the discussion of the roles for LLMs (interlocuter, monitor, and respondent) to be interesting. I think this framing is interesting to consider in an educational context (e.g., how should universities position themselves with relation to AI assistants). Further the paper is well written and does a solid job of situating the current use cases of LLMs in contrast with their use in fostering critical thinking.

**Reasons To Reject:**

The link between the paper’s aim (positioning LLMs as critical thinking tools) and the argumentation and methods (the interview of academic philosophers) is not clear. If the aim is to study what makes a tool useful for supporting critical thinking, why are philosopher’s views on LMs (e.g., answers to questions like “What are some risks and weaknesses for language models in philosophy”) relevant for addressing this? At times the paper discusses how critical thinking is deployed in endeavors like “history, science, and philosophy”, wouldn’t it make sense to probe how each of these fields utilizes critical thinking and what tools are useful for doing this more efficiently?

Further, the discussion of the things that are useful for philosophers in critical thinking, selfhood and initiative, both require agency (in my understanding) in the definitions given in the paper (“selfhood is a resource’s ability to have certain locally persistent internal states (such as perspectives, beliefs, opinions, memory) and to consistently use them as the basis for judgements” and “Initiative is a resource’s ability to set its own intentions and goals, possibly different from its user’s, and to execute actions oriented towards those intentions.”). This feels not concrete enough to offer practical guidance in developing LLMs.

---

> ### Author Rebuttal · Authors · 2024-05-28
>
> We appreciate the reviewer’s feedback! To address the reasons to reject (RR) and questions:
>
> [Q1]  To clarify: we outline in the intro and the related work that LLMs are often used as tools to automate some types of thinking (e.g. coding, emails), but that people want to be the ones doing the thinking when they’re engaging in critical thinking tasks; AI tools should be augmentative here, not automative.
>
> [Q2] The reviewer suggests that the assertion that LMs lack selfhood and initiative is not empirically supported. Indeed, our paper is not quantitatively empirical; rather, we interview and systematically extract themes from philosophers’ experiences with LMs. The claims that LMs are neutral, detached, etc. are directly taken from the interviews with interviewee citations and quotes. LM scientists are finding now that some of the most useful metrics come not from static benchmarks but from real users (e.g. LMSYS Arena), often experts (see reviewer NJTN). Our paper presents a detailed, qualitatively empirical evaluation of a set of critical thinking experts’ (philosophers) experiences with current LMs as critical thinking tools.
>
> [RR1] The reviewer suggests the link between critical thinking tools and the choice to interview philosophers is unclear. To clarify, we specify in S2.3 that we study philosophers as a case study of how people might use LMs as critical thinking tools, as nearly all philosophical problems require critical thinking. We discuss how critical thinking is used in many fields in the intro, but narrow in on philosophy for the paper – so we believe we do “probe how [one] of these fields utilizes critical thinking and what tools are useful” as the reviewer suggests. In order to do this probing, we need to holistically understand philosophers’ experiences with LMs (such as weaknesses of LMs for philosophy).
>
> [RR2] Selfhood and initiative can certainly be thought of as reflecting aspects of agency, which we agree is a fraught & abstract concept. However, we believe we have presented selfhood and initiative in concrete ways which do not require addressing agency. Many attributes of selfhood and initiative are already LM research areas, such as using memory / persistent states, user-specific alignment, autonomous planning and tool use, theory of mind, etc. We describe and cite several such works in S6.2. Our work draws attention to certain existing (but possibly underappreciated) problems as important for LM critical thinking tools.
>
> Thank you!

---

> > ### Comment · Reviewer_Ynvx · 2024-06-05
> >
> > Thanks for the engagement with the review.
> >
> > [Q1] Thank you for the clarification.
> >
> > [Q2] I think the evaluation lacked some detail. It is unclear how often the interviewed people used LLMs, in what domains they used them, and which ones they used. It does not seem that any of the questions in A.1 ask for how participants use or have used LLMs for critical thinking (the responses to Reviewer veW6 further add confusion to this. In what ways were interviewees new to LLMs interacting with LLMs, for example?). It would be interesting to get a sense of this and how it relates to their responses, for example.
> >
> > [RR1] Thanks for the clarification.
> >
> > [RR2] I don't see how the definitions given do not require agency, given the quoted definitions from the paper I included in the review.
> >
> > Overall, I still find the paper flawed and the main argumentation unclear. If the aim is to do a study like those in HCI work, the survey is lacking details about the users and their experiences which limit the inferences we can draw from their study. I will keep my score.

---

### Official Review · Reviewer_ttPJ · 2024-05-15

**Rating:** 7
**Confidence:** 3
**Ethics Flag:** 1

**Summary:**

The paper discusses the use (potential and actual) of LMs by philosophers as critical thinking tools. From interviews with philosophers, the authors determine two broad groups of reasons why philosophers find current LMs problematic as thinking tools. Based on this, they propose a basic framework to characterise such tools, and suggest how LM outputs and behaviour might be designed to give better support for this kind of use.

**Questions To Authors:**

The two-dimensional self/initiative model for critical thinking tools is proposed without a great deal of justification. While I can see that it links to the two main themes identified from the analysis of the interviews, I wasn't clear if it has any precedent or builds on / relates to other existing models of critical thinking or of creative assistive technology. There is plenty of prior work in designing tools to assist creative thinkers (artists, designers etc - I know of work dating back at least to Shneiderman (1997)'s Genex framework but I'm no expert in that area so I'm sure there's more), and there seems to be work in categorising critical thinking according to various dimensions. Does this model relate to previous work, and if so how? This would help justify this choice.

**Reasons To Accept:**

This is a thought-provoking and original piece of work: it's useful to see discussion of the interaction design aspects of LMs, and serious consideration of how they might support (or not) different kinds of end use.

**Reasons To Reject:**

The findings about reasons why LMs are not much used in this context are interesting and seem solid, but the analytical framework proposed to explain this and used as the basis for suggestions does not seem to be so clearly justified; and the suggestions are fairly open challenges without much in the way of suggestions as to how (or whether) they could be implemented or addressed.

---

> ### Author Rebuttal · Authors · 2024-05-28
>
> We appreciate the reviewer’s feedback!
>
> We agree with the reviewer that there could be more contextualization of the selfhood-initiative (SI) model. The SI model comes directly from our interviews, in which we discovered two central themes in philosophers’ experiences with LMs: LMs are “neutral, detached, and nonjudgmental” (lack selfhood), and “servile, passive, and incurious” (lack initiative). These themes serve as the axes in our model, which we use to identify roles in which at least one of the axes is positive.
>
> Our model is also related to other work from a variety of fields:
>
> - Many recent HCI works are related to the “initiative” axis of our model. Liao et al. (“Nudge for Reflective Mind”, 2022), Danry et al. ("Don’t Just Tell Me, Ask Me", 2023), inter alia, discuss how building proactively prompted moments for user reflection improve users’ critical thinking.
> - Many education studies emphasize grounded conversation for developing critical thinking (“selfhood” axis). Lang (“Critical thinking in web courses”, 2000) writes that “critical thinking happens in good discussions”. Students have such good discussions when they actively engage with other students’ ideas (Bai, “Facilitating Students’ Critical Thinking in Online Discussion”, 2009), such as through role-play (Schindler et al., “Instructional Design and Facilitation Approaches”, 2014).
>
> However, to our knowledge, our model is the first to cohesively synthesize these two axes and relate it to the design of LMs.
>
> W.r.t. “The suggestions are fairly open challenges…” We believe that the way in which we have described selfhood and initiative are concretely linked to existing LM research problems, such as in LM memory use, user-specific alignment, autonomous planning and tool use, theory of mind, etc. (More in S6.2.) We want to draw attention to certain existing (but possibly underappreciated) challenges as important for making LMs useful as critical thinking tools. For instance, researchers might build LM systems which (safely) perform multiple uninterrupted steps of conversation (e.g. asking questions, using tools, changing topics) without human supervision. Alternatively, researchers might develop methods for LMs to predict more advanced elements in theory of mind, such as uncovering implicit values and unspoken confusions (by anticipating questions users might ask), or even begin by building a more complex taxonomy and benchmark.
>
> Thank you!

---

### Decision · Program_Chairs · 2024-07-10

**Decision:**

Accept

**Comment:**

This paper has a significant spread of review scores that was not resolved by discussion or rebuttal.

The paper under review explores the use of language models (LMs) by philosophers as tools for critical thinking. It is based on interviews with 21 professional philosophers, who were asked to reflect on the utility of LMs in their work. The authors identify two primary reasons why philosophers currently find LMs problematic: the lack of selfhood and initiative. They propose a framework to characterize these tools and suggest improvements to LM outputs and behaviors to better support critical thinking.

Reasons to Accept
====
The paper presents a novel perspective by discussing the interaction design aspects of LMs and how they might support critical thinking, an area not extensively covered in existing literature.
* The qualitative study involving interviews with professional philosophers provides valuable empirical data on the attitudes towards and the perceived limitations of LMs.
* The proposed "selfhood-initiative" model offers a new conceptual framework to understand and enhance the role of LMs in critical thinking.
* The insights and proposals could significantly impact interdisciplinary researchers working with LMs, contributing to the broader discourse on AI and philosophy.

Reasons to Reject
====
* The justification for the "selfhood-initiative" model is not well-developed. The connection between this model and existing theories or frameworks in critical thinking or creative assistive technology needs to be clarified.
* The relevance of philosophers' views on LMs to the broader aim of studying tools for critical thinking across various fields (history, science, philosophy) is not well-argued. Additionally, the criteria for selecting interviewees and the specific LMs used in the study are not detailed. Although


Despite some methodological and theoretical weaknesses, the empirical insights and conceptual contributions make it a valuable piece of work. It addresses an underexplored area at the intersection of AI and philosophy, offering a basis for future research and development.

[comment from the PCs] Please make sure to add the details as noted missing by the AC.